# Diabetes and Familial Hypercholesterolemia: Interplay between Lipid and Glucose Metabolism

**DOI:** 10.3390/nu14071503

**Published:** 2022-04-03

**Authors:** Ana M. González-Lleó, Rosa María Sánchez-Hernández, Mauro Boronat, Ana M. Wägner

**Affiliations:** 1Endocrinology and Nutrition Department, Complejo Hospitalario Universitario Insular Materno-Infantil, 35016 Las Palmas de Gran Canaria, Spain; agonlle@gobiernodecanarias.org (A.M.G.-L.); mborcor@gobiernodecanarias.org (M.B.); 2Instituto Universitario de Investigaciones Biomédicas y Sanitarias, Universidad de Las Palmas de Gran Canaria, 35016 Las Palmas de Gran Canaria, Spain

**Keywords:** familial hypercholesterolemia, diabetes, LDL receptor, genetic risk, insulin resistance, review

## Abstract

Familial hypercholesterolemia (FH) is a genetic disease characterized by high low-density lipoprotein (LDL) cholesterol (LDL-c) concentrations that increase cardiovascular risk and cause premature death. The most frequent cause of the disease is a mutation in the LDL receptor (*LDLR*) gene. Diabetes is also associated with an increased risk of cardiovascular disease and mortality. People with FH seem to be protected from developing diabetes, whereas cholesterol-lowering treatments such as statins are associated with an increased risk of the disease. One of the hypotheses to explain this is based on the toxicity of LDL particles on insulin-secreting pancreatic β-cells, and their uptake by the latter, mediated by the LDLR. A healthy lifestyle and a relatively low body mass index in people with FH have also been proposed as explanations. Its association with superimposed diabetes modifies the phenotype of FH, both regarding the lipid profile and cardiovascular risk. However, findings regarding the association and interplay between these two diseases are conflicting. The present review summarizes the existing evidence and discusses knowledge gaps on the matter.

## 1. Introduction

### 1.1. Familial Hypercholesterolemia

Familial hypercholesterolemia (FH) is a genetic disease characterized by high low-density lipoprotein (LDL) cholesterol (LDL-c) concentrations that increase cardiovascular risk and cause premature death [1]. The most frequent mutations are found in the LDL receptor gene (-*LDLR*- responsible for LDL uptake), though other genes involved in LDL metabolism can also cause the disease, such as apolipoprotein B 100 (*APOB*), apolipoprotein E (*APOE*) or proprotein convertase subtilisin/Kexin-type 9 (*PCSK9*) [2,3]. Heterozygous FH (HeFH) (one affected allele) is the usual presentation form, with a prevalence of 1/250 [4], higher in isolated regions [5,6,7]. LDL-c concentrations in people with HeFH are often twice those of the general population [8]. Homozygous FH (HoFH) is infrequent (1/160,000–1/300,000) but more severe, with LDL-c concentrations exceeding 500 mg/dL from birth. Without treatment, subjects with HoFH develop atherosclerosis before the age of 20 and die before 30 [9]. The diagnosis of FH is usually made based on LDL-c concentrations, family history, and the presence of corneal arcus, xanthomas, or xanthelasmas [8]. Although affected individuals have a higher cardiovascular risk than the general population [10], subjects with the same mutation show enormous phenotype variability. These differences might be explained by other factors such as the type of mutation [11], age [12], gender [10,13], or the existence of other concomitant diseases [14].

### 1.2. Diabetes Mellitus

Diabetes mellitus (DM) is a group of metabolic disorders defined by increased blood glucose concentrations. The most frequent types of DM are type 1 diabetes (-T1DM- mediated by autoimmune destruction of pancreatic ß cells and absolute insulin deficiency), type 2 diabetes (-T2DM- caused by progressive loss of insulin secretion in the context of insulin resistance) and gestational DM (first diagnosed during pregnancy), but there are also other, less frequent forms of the disease, such as monogenic DM or drug-induced DM [15]. A correct classification of DM is important since both treatment and follow-up depend on it. The prevalence of DM has doubled since the 1990s [16]; nowadays, there are about 537 million subjects with DM around the world (mostly T2DM), and this is expected to continue increasing in the near future [17]. Its complex physiopathology involves modifiable factors such as weight, diet, or physical activity [18], and non-modifiable factors such as genetics, age, or gender [19]. Patients have an increased all-cause mortality [20], but about 50% die because of cardiovascular complications [21], especially women [22], and people with long-standing disease [23,24]. This cardiovascular risk is enhanced in the presence of other risk factors such as smoking, hypertension, or dyslipidemia that contribute to endothelial damage and the progression of atherosclerosis [25].

The prevalence of DM is generally lower in people with FH than in the general population [26], suggesting a relationship between glucose and lipid metabolism. The aim of this paper is to summarize the existing evidence and contribute to the understanding of the complex underlying mechanisms that relate DM and HF.

## 2. Familial Hypercholesterolemia and Diabetes: Molecular Causes

### 2.1. Genetics of FH

FH is the most common monogenic disorder. It has high penetrance (90%) and autosomal dominant inheritance [1] and is caused by mutations in genes related to LDL metabolism.

HeFH is mainly caused by loss-of-function mutations in *LDLR* (85–90%) or *APOB* (5%), or gain-of-function mutations in *PCSK9* (1–3%) [27]. Mutations have also been identified in *APOE* [3] and in the adaptor protein type 1 gene (*LDLRAP1*), the latter with autosomal recessive inheritance [28]. However, 10–40% of patients with a clinical phenotype of FH have negative genetic tests, probably representing severe polygenic forms of hypercholesterolemia [29].

HoFH is a more severe form that involves two mutations in the aforementioned genes. According to the combination of mutations, HoFH is classified into the following: true homozygotes (two equal mutations in both alleles of the same gene, mostly in *LDLR)*; compound heterozygotes (a different mutation in each allele of the same gene); double heterozygotes (two different mutations in different genes); autosomal recessive hypercholesterolemia (mutations in *LDLRAP1*) [9]. The phenotype of HFHo will depend on the degree of residual LDLR activity, which is defined by the genetic defect. Indeed, in some cases, the LDLR protein is almost absent (less than 2%), leading to the most extreme phenotypes [30].

*LDLR* is the most frequently affected gene in HF and more than 3000 mutations have been described so far, most of them disease-causing or pathogenic [2]. Traditionally, mutations were classified into classes I to V, with class I mutations being the most severe, where no protein synthesis is present (large rearrangements, insertions, nonsense frameshifts, or splicing mutations). Classes II-IV include alterations in LDLR transport, LDLR binding, internalization, or recycling of LDLR, corresponding to in-frame, missense mutations, or small deletions [27]. Currently, there is a tendency to simplify this classification into class 1 and non-class 1 mutations [31], which would correspond to null or defective alleles, respectively, and this correlates with the severity of the individual phenotype. Null *LDLR* allele carriers present with very high LDL-c concentrations, premature coronary heart disease and poor response to treatment [32]. However, LDL-c concentrations have been shown to improve cardiovascular risk prediction more than the genetic defect per se. A cohort study in 12,245 FH *LDLR* mutation carriers showed that the classification of pathogenic *LDLR* variants according to LDL-c concentration percentile was indeed more accurate than class 1 vs. non-class 1. The relative risk of major cardiovascular events ranged from 2.2 in subjects with an LDL-c concentration below the 75th percentile to 13 when the LDL-c concentration was above the 98th percentile of the cohort [33].

*APOB* was the second gene identified to be associated with FH, also called familial defective APOB [34]. It is less frequent than FH caused by *LDLR* mutations, and there are currently about 35 pathogenic mutations described, generally located in the LDLR-binding domain of apolipoprotein B (apoB) [27]. The most common is the R3500Q mutation, which accounts for 5–10% of FH cases in northern Europe [35]. Patients with this form of FH present with less severe phenotypes than *LDLR* mutation carriers and have lower LDL-c concentrations and less cardiovascular events [36].

FH type 3 is caused by gain of function mutations in *PCSK9* [37], and there are about 30 pathogenic variants reported [27]. The phenotype is variable, with variants such as p. (Asp374Tyr), which causes an extreme FH phenotype with very high LDL-c concentrations and premature coronary heart disease [38], and other mutations affecting distinct domains of the protein, leading to milder phenotypes and better response to treatment [39].

In patients with an FH phenotype but no mutation identified, a polygenic mechanism should be considered, caused by the aggregation of common LDL-c-raising genetic variants or single nucleotide polymorphisms (SNPs), which can be studied using validated polygenic risk scores [40,41].

There are other genes that are no longer considered to cause FH, such as *STAP1*, which seemed to be associated with the disease, but subsequent in vitro and family segregation studies have shown that it does not cause FH [42,43].

### 2.2. Genetics of Type 2 Diabetes

Regarding the genetics of DM, there are both monogenic forms, including neonatal diabetes mellitus and maturity-onset diabetes of the young (MODY), and the following polygenic forms: T1DM or T2DM [44]. Neonatal diabetes is caused mainly by paternally inherited duplications in chromosome 6q24 that cause overexpression of paternally imprinted genes, mutations in K_ATP_ channels, potassium inwardly rectifying channels, subfamily J, member 11 (*KCNJ11*) or ATP Binding Cassette Subfamily C Member 8 (*ABCC8*) genes, among others [45]. Mutations in the hepatocyte nuclear factor 1-α (*HNF1A*), 4-α (*HNF4A*), 1-ß (*HNF1B/TCF2*) and glucokinase (*GCK*) genes are responsible for most of the cases of MODY [46].

The development of T2DM depends on both environmental [47] and genetic causes. The genetics of T2DM are very complex, and genome-wide association studies and whole-genome sequencing have shown more than seventy genes related to the pathogenesis of the disease [48,49]. A large number of SNPs have been described in more than 400 distinct genomic regions [50]. The heritability of T2DM ranges from 20 to 80% [51], the highest concordance corresponding to monozygotic twins [52]. Despite the huge number of risk SNPs identified, each one accounts only for a small effect on the risk of T2DM, around 10–20% increase per risk allele [44]. Because of this, various genetic risk scores have been developed to evaluate the cumulative effect of multiple SNPs and to identify individuals with a high genetic risk of T2DM [53,54].

The genes with the most reported risk variants are *KCNJ11*, peroxisome proliferator-activated receptor gamma (*PPARG*), *HNF1B/TCF2* and wolfram syndrome 1 (wolframin) (*WFS1*), confirmed by genome-wide association studies [55]. Other genes related to T2DM are insulin receptor substrate 1 gene (*IRS1*) and *IRS-2*, *ABCC8*, Phosphatase and Tensin Homolog (*PTEN*), Zinc Transporter-8 Gene (*SLC30A8*), GATA Binding Protein 6 (*GATA6*), ISL LIM Homeobox 1 (*ISL-1*), Transcription Factor 7-like 2 (*TCF7L2*), Insulin-like Growth Factor 2 mRNA-Binding Protein 2 (*IGF2BP2*), among many others [48,50,56].

The effects of variants in these genes can lead to impaired insulin response, decreasing insulin sensitivity, loss of the ß cell morphology, generate oxidative stress in the pancreas, destruction of pancreatic β-cells altering insulin biosynthesis, causing insulin receptor dysfunction, etc. [48,56]. Due to the polygenic feature, many genes and their SNPs contribute to an enhanced risk of T2DM, which together with environmental triggers, like obesity, leads to the development of the disease [51].

### 2.3. Genetic Studies Assessing the Link between Hyperlipidemia and Type 2 Diabetes

Mendelian randomization studies suggest that there is an overlap between the risks of DM and hyperlipidemia. Indeed, after combining and analysing existing information provided by three large consortia, Fall et al. report a significant association between gene variants determining higher LDL-c and a lower risk of T2DM, whereas the association with variants determining HDL-c and triglycerides was less clear [57,58]. When constructing the risk scores, the authors excluded SNPs associated with adiposity, which they considered a possible confounder. White et al. used a modified approach in a dataset combining several genome-wide association studies, including 188,577 individuals with measured blood lipids and 34,840 with T2DM. A 130 SNP score was developed for LDL-c (explaining 7.9% of its variance), and 140 SNP scores, for HDL-c and triglycerides. For each SD (38 mg/dL) estimated increase in LDL-c, the risk of T2DM was reduced by 21% (R 0.79 (0.71–0.88)). For triglycerides, every 89 mg/dL estimated increase was also associated with a reduction in T2DM (OR 0.83 (0.72–0.95)), as was the case for every 16 mg/dL estimated increase in HDL-c (OR 0.83 (0.76–0.90)) [59]. Although the protective effect of triglycerides seems somewhat unexpected, other studies in different ethnic groups agree with this finding [60,61].

## 3. Familial Hypercholesterolemia and Glucose Metabolism: Risk of Diabetes

### 3.1. Epidemiological Studies

In 2019, the worldwide prevalence of DM was 9.3%, higher in men (9.6 vs. 9%) and in high-income countries (10.4 vs. 4%) [17]. Most epidemiological studies in FH subjects have shown a lower DM prevalence than in the general population (see Table 1). In a Dutch cohort with more than 14,000 FH subjects, only 2.8% had DM [62], whereas a British cohort showed an even lower prevalence (0.8%) [63], and intermediate results were described in 263 French-Canadian patients with FH [64]. Recently, a Spanish study with more than 1700 subjects with FH found a T2DM prevalence close to 6%, around one third of the national average [65]. However, another recently published Spanish study, performed on the island of Gran Canaria, showed an unexpectedly high prevalence of DM in HeFH *LDLR* mutation carriers (25%) [66]. Other studies show a high prevalence of DM too, above 20%, but in patients with only clinical diagnosis of FH without genetic confirmation [67,68].

Regarding the relationship between FH mutations and DM, the results are not consistent. Patients with mutations in *APOB,* with a less severe phenotype, had a higher prevalence of T2DM (1.91%) than *LDLR* mutation carriers, and amongst these, the most severe phenotype (receptor-negative) had the lowest prevalence of DM (1.12%) [26]. In accordance with these findings, *PCSK9* InsLEU mutation carriers had a higher prevalence of DM and a lower incidence of coronary heart disease. However, other studies have not found an association between mutation type and DM [74,75].

### 3.2. Lipid-Lowering Treatment and Risk of Diabetes

In recent years, many drugs have been developed to treat hypercholesterolemia, and several studies have shown that they could alter glucose tolerance, highlighting the link between cholesterol and glucose metabolism (see Table 2).

#### 3.2.1. Statins

Statins are the treatment of choice for hypercholesterolemia, both in primary and secondary prevention [93,94]. They inhibit the 3-hydroxy-3-methylglutaryl-coenzyme A reductase (HMG-CoA reductase), increase LDLR expression, and reduce plasma LDL-c concentration by over 50% [95]. New-onset DM (NODM) has a prevalence of 9–12% and is one of most recognized side effects of statins [76,96]. Risk increases with age in women [97], and in people with more than two risk factors for DM (impaired fasting plasma glucose, hypertriglyceridemia, hypertension, obesity, or the metabolic syndrome) [77,78]. The risk of DM seems to be independent of LDL-c concentrations [76,79] and varies according to statin type and dose, as well as exposure time [80,98]. Nevertheless, this association with NODM should not discourage health professionals from prescribing these drugs, given their proven cardiovascular benefit, especially in high-risk individuals [99,100]. Simvastatin, atorvastatin, and rosuvastatin have shown more glucose impairment, while pitavastatin has a lower risk of NODM compared with atorvastatin and rosuvastatin [81,96,101]. Pravastatin has also shown favourable results, probably related to its lower liposolubility and limited potency [82]. However, FH subjects seem to be protected against these diabetogenic effects [70].

#### 3.2.2. Ezetimibe

Ezetimibe inhibits intestinal absorption of cholesterol by blocking the Niemann-Pick C1 like1 (NPC1L1) transporter [102], and is frequently used as a concomitant treatment to statins. Its relationship with glucose metabolism is controversial. Several studies have shown that fasting plasma glucose, glycosylated haemoglobin (HbA1c) and insulin sensitivity improve with ezetimibe treatment, both in DM and non-DM individuals [103,104]. This drug also improves inflammation markers and obesity and reduces waist circumference [83]. Based on these positive results, a possible compensatory effect on the diabetogenic effects of statins has been studied. Dragi et al. found that the combination of low-dose-pravastatin plus ezetimibe improved insulin resistance and inflammation compared with high-dose-pravastatin alone [84]. In 2018, a meta-analysis concluded that patients who used low-dose-statins plus ezetimibe for more than 3 months had lower fasting plasma glucose compared with those treated with high-dose statins [105]. Nevertheless, no differences in the HOMA-IR index were found when two statins in monotherapy were compared with a combination of low-dose-statin plus ezetimibe [85]. No significant differences were found either, in a recent study that compared statins alone versus their combination with ezetimibe in glucose intolerant patients followed for 7 years [106]. Other studies have found neutral [107] or deleterious effects on glycemic metabolism with ezetimibe, with an increase in HbA1c and hepatic long-chain fatty acids in patients with non-alcoholic fatty liver disease [86].

The discrepancies in the results could be explained by the small number of participants in some studies, insufficient follow-up, or the presence of other lipid-lowering drugs that could act as confounders.

#### 3.2.3. PCSK9 Inhibitors (PCSK9-i)

Inhibition of the PCSK9 enzyme prevents LDLR degradation after cellular internalization, reducing LDL-c by about 60%. Approved in 2015, monoclonal antibodies against PCSK9 (alirocumab and evolocumab) have shown a favourable safety profile with few side effects [108], but the consequences on glucose metabolism are still not clear. Despite the fact that most clinical studies have not found an association between PCSK9-i and NODM or worsening of pre-existing DM [87,109].

A large study including more than 96,000 individuals followed for 1.5 years found a small but significant increase in plasma glucose and HbA1c but not a higher incidence of NODM in those treated with PCSK9-i [88]. In 2020, a meta-analysis found that alirocumab was associated with a reduction in the risk of DM and, when compared with ezetimibe in monotherapy, evolocumab was also associated with this risk reduction. However, when used in combination with statins, an increased risk of NODM was found in the PCSK9-i group, even though the use of statins was equivalent between the experimental and active comparator arms [89]. It seems that the combination with other lipid-lowering drugs (especially statins) could change the studies’ results due to the discrepancies in background treatment between groups. Furthermore, mendelian randomization studies must be interpreted carefully. As is the case for other lipid-lowering drugs, follow-up is often limited and could be insufficient to see an effect on glucose metabolism [110].

#### 3.2.4. Bempedoic Acid

Bempedoic acid is a newly developed drug that inhibits adenosine triphosphate citrate lyase, increasing LDLR expression and reducing LDL-c [90]. In the phase 3 “CLEAR” studies, bempedoic acid was associated with a reduced incidence of DM and an improvement in fasting blood glucose and HbA1c in week 12 in pre-DM or DM subjects, without increasing NODM risk for 1 year [90,91]. A recent meta-analysis found a reduction of 34% in NODM risk [91].

#### 3.2.5. Other Cholesterol-Lowering Drugs

Nicotinic acid (B3 vitamin) reduces triglyceride and LDL-c concentrations and raises high-density lipoprotein cholesterol (HDL-c) by up to 35% [111]. It is associated with an increased risk of NODM and higher fasting plasma glucose and HbA1c, especially in predisposed individuals, with a dose-dependent effect [112]. Niacin has other side effects, such as flushing, and does not reduce cardiovascular events in secondary prevention [113], so its use is currently limited.

Bile acid sequestrants (resins) reduce bile acid reabsorption and increase hepatic LDLR, lowering LDL-c by 15–25%. They improve the glucose profile but do not cause hypoglycemia in T2DM subjects. Similar results have been found with different resins and in both pre-DM and healthy individuals [92,112,114]. Although they have a moderate lipid-lowering effect, they could be useful in subjects with DM because of their dual effects on lipid and glucose metabolism.

### 3.3. Genetics and Metabolism

The cause of the lower prevalence of DM in FH subjects found in most studies is not clearly known yet. In vitro, long exposure to fatty acids has been associated to β-cells dysfunction and reduced insulin secretion, especially when coexisting with hyperglycemia [115,116]. Moreover, in vitro studies have shown that intracellular cholesterol accumulation also induces apoptosis of pancreatic β-cells [117]. LDL particle uptake causes β-cell death in a dose-dependent manner, and this toxicity can be counteracted by HDL, very LDL (VLDL) particles, or antioxidants [118]. Supporting these findings, polymorphisms in ATP-binding cassette transporter 1 gene (*ABCA1*), involved in cholesterol efflux and HDL synthesis, have been associated to obesity, the metabolic syndrome, and DM [119,120]. On the β-cell, HDL particles have an anti-inflammatory effect and participate in cholesterol efflux [121]. Higher HDL-c levels are associated with less hyperglycemia and HDL particle size is inversely correlated to T2DM risk in the general population [122].

A large meta-analysis of genetic association studies assessing the effects of cholesterol-lowering variants in or near *NPC1L1*, *HMGCR*, *PCSK9*, *ABCG5*/G8 and *LDLR* showed an overall increased risk of DM with an odds ratio of 1.19–2.42 for every 1 mmol/L (38.6 mg/dL) reduction in LDLc [110]. However, there was rather high heterogeneity in the meta-analysis, suggesting gene-specific associations with DM. Indeed, the highest risk of T2DM was associated with variants in or near *NPC1L1*, whereas the *HMGCR* locus was associated with body mass index and waist-to-hip ratio, and *PCSK9,* with higher fasting and two-hour glucose concentrations [110].

The lipotoxicity hypothesis could, at least partially, explain how statins increase NODM and how FH reduces the risk of DM. The rise in LDLR increases LDL particle uptake by pancreatic β-cells, thereby promoting dysfunction and apoptosis, especially in those with baseline glucose disturbances. On the other hand, genetic mutations that prevent cholesterol input, like FH, could be protective and explain the inverse relationship between mutation severity and DM prevalence [123]. However, clinical studies do not clearly reflect this theory. No differences in insulin, C peptide, or fasting plasma glucose concentrations have been found comparing FH with non-FH subjects, regardless of their insulin sensitivity [124,125,126]. Indeed, in some studies, FH has even been associated with an increased risk of impaired glucose metabolism [7,127].

In vivo studies show controversial results. When comparing prediabetic wildtype vs. LDLR knock-out (KO) mice, no differences were observed in glucose levels, although less insulin secretion and more β-cell apoptosis were seen in LDLR KO mice [128].

In a study in PCSK9 KO and PCSK9/LDLR double knock-out mice, the former showed reduced insulin secretion and glucose intolerance, as well as cholesteryl ester accumulation in β-cells compared with WT mice. In the double knock-out mice, these alterations were restored, supporting the hypothesis that LDLR, the target of PCSK9, is responsible for the phenotype [129]. However, a later study with PCSK9 KO and PCSK9 ß-cell specific KO mice does not show any alteration on glucose homeostasis nor in β-cell function [130].

Thus, other molecular or environmental factors are probably involved in DM risk. For example, plasma lipoprotein(a) (Lp(a)) has been shown to be higher in HeFH compared with the general population [131], and an inverse association has been described between Lp(a) concentrations and the risk of T2DM [132]. However, this effect has to be confirmed, and a mechanism explaining it is still to be found.

Regarding environmental factors, a study comparing a cohort of 2185 HeFH subjects from the Spanish Dyslipidaemia Registry with a representative sample of the background population showed more favorable cardiovascular risk profiles in the former. Indeed, HeFH subjects without cardiovascular disease showed a lower body mass index and a lower prevalence of smoking than the background populations, suggesting that the lower prevalence of T2DM could, at least partially, be explained by a healthier lifestyle in patients with FH [133].

## 4. Coexistence of Diabetes and Familial Hypercholesterolemia: Clinical Consequences

### 4.1. Effects on the Lipoproteins

Cardiovascular disease is the leading cause of death in people with DM. Traditionally, DM has been considered to increase the risk of ischemic heart disease, stroke, and peripheral arterial disease by 2–4 times [134]. Although recent studies show that contemporary treatment for cardiovascular risk has reduced the excess mortality associated with the disease, DM remains a very strong independent risk factor for cardiovascular morbidity and mortality [135]. Therefore, since FH is associated with an elevated risk of premature atherosclerosis, it is conceptually reasonable to assume that the coexistence of both DM and FH has a strong impact on cardiovascular disease risk.

While decreased clearance of LDL particles and accumulation of LDL-c is the main determinant for increased cardiovascular disease in FH, multiple interconnected mechanisms have been involved in vascular damage caused by DM, including hyperglycemia-induced overproduction of reactive oxygen species, accumulation of advanced glycation products, activation of protein kinase C and chronic inflammation [136]. In addition, DM is also responsible for a characteristic cluster of lipid disorders with high atherogenic potential, known as diabetic dyslipidemia. Although diabetic dyslipidemia and FH share hyperbetalipoproteinemia as the fundamental mechanism for atherogenesis, the mechanisms behind them and their biochemical expression are different.

The hallmarks of diabetic dyslipidemia are hypertriglyceridemia and decreased HDL-c, whereas LDL-c concentrations are normal or only slightly increased. Although the mechanisms of diabetic dyslipidemia are not completely understood, it is accepted that insulin resistance is its main underlying element [137]. Under physiological conditions, insulin inhibits lipolysis in adipose tissue and activates lipoprotein lipase, an enzyme involved in the plasma clearance of triglycerides from VLDL and chylomicrons. In a state of insulin resistance, lipolysis is not inhibited, and increased circulating free fatty acids are readily taken up by the liver and used as substrates for synthesis and subsequent release of VLDL. Hypertriglyceridemia stimulates the enzymatic activity of cholesteryl ester transfer protein and, during their passage through the circulation, VLDL particles transfer their triglycerides to HDL and LDL in exchange for cholesteryl esters [137]. Triglyceride-enriched HDL undergoes lysis by hepatic lipase, a mechanism by which they are converted into small, dense particles with reduced antioxidant, anti-inflammatory, and anti-atherogenic capacity compared to normal HDL. The smaller HDLs, in turn, are cleared more rapidly from the circulation, resulting in a decrease in HDL-c and apolipoprotein A-1 (apoA-1) concentrations [137]. In a similar manner, LDL particles also become smaller and denser due to a higher ratio of protein to lipid (LDL phenotype B). These LDL particles are resistant to receptor binding, pass more readily through the arterial wall, bind to proteoglycans and are more susceptible to oxidation [138]. On the whole, although LDL-c is not characteristically increased, diabetic dyslipidemia is characterised by an increase in the total number of apoB-containing particles (VLDL, IDL, and LDL).

Several studies have assessed the presence of phenotypic features of diabetic dyslipidemia in non-diabetic subjects with FH. LDL particles from both HoFH and HeFH patients appear to be larger, more buoyant, and more resistant to oxidation than those from healthy controls [139]. Thus, the qualitative properties of LDL do not seem to play a significant role in the development of atherosclerosis in people with FH. Furthermore, patients with FH usually have normal triglyceride concentrations. However, experimental studies have suggested that defective LDLR promotes liver uptake of chylomicrons and remnants and increases VLDL secretion [140,141]. In fact, disturbed triglyceride-rich lipoprotein metabolism and, particularly, postprandial dyslipoproteinemia have been proposed as a putative modulator of cardiovascular risk in HeFH [142]. The possible role of lipoprotein lipase in postprandial hyperlipemia among subjects with HeFH has not been specifically studied. However, individuals with HeFH who carry an *LPL* gene variant that reduces lipoprotein lipase activity, show higher triglyceride levels and lower HDL-c levels than non-carriers of this mutation [143]. This suggests that a decreased lipoprotein lipase activity, as occurs in insulin resistance, could condition the phenotype of HeFH. Finally, results have been discordant regarding serum concentrations of HDL-c in subjects with FH [141]. This is probably related to the fact that, in subjects with FH, there is an increase in both synthesis and catabolism of HDL particles, but there may be an imbalance between both processes that varies depending on population-specific genetic or environmental factors. Increased apoA-1 catabolism due to increased cholesteryl ester transfer protein activity favours the generation of small HDL particles rich in triglycerides and apolipoprotein E [144,145]. Moreover, HDL particles in subjects with FH may show different functional abnormalities not detectable by measuring HDL-c alone. This may include a defective ability to reverse cholesterol transport from macrophages and impaired anti-inflammatory and antioxidant capacity [144,145].

As mentioned above and depicted in Figure 1, it is reasonable to think that subjects with FH who develop DM may have alterations in lipid metabolism resulting from the additive effect of both diseases. A few studies have compared the clinical characteristics and lipid profiles of HeFH subjects with and without T2DM [68,74,146]. Patients with DM were older, had a higher prevalence of hypertension, and had a higher body mass index than patients without DM. As expected, they also had a lipid profile more characteristic of diabetic dyslipidemia, including higher triglyceride and lower HDL-c and apoA-1 concentrations [68,74,146], as well as higher concentrations of markers of subclinical systemic inflammation, such as C-reactive protein and neutrophil count [68], typical of individuals with insulin resistance.

### 4.2. Effects on Chronic Arterial Wall Inflammation and Endothelial Dysfunction

In recent decades, abundant scientific evidence has highlighted the preponderant role of immunological and inflammatory mechanisms in the development and progression of atherosclerosis. As mentioned above, inflammatory mechanisms may be particularly important in the development of cardiovascular disease in individuals with T2DM. Epidemiological studies have shown that insulin resistance is associated with high concentrations of uric acid and a wide set of acute phase reactants and markers of endothelial dysfunction [147,148]. In addition, obesity, commonly present among people with T2DM, perpetuates the maintenance of a state of chronic inflammation as adipose tissue secretes a variety of proinflammatory adipocytokines such as tumour necrosis factor α, interleukins 1, 6, and 8, resistin, adiponectin, leptin, and adipsin [149].

Increased blood concentrations of different biomarkers of systemic inflammation, endothelial activation, and oxidative stress [150,151] have also been reported in FH subjects, and some authors have postulated their possible role as tools for cardiovascular risk stratification in HeFH [152]. In any case, these studies reveal that DM and FH could share a greater predisposition to the activation of pathways leading to arterial wall inflammation and endothelial activation, promoting early mechanisms of atherosclerosis induction.

### 4.3. Effects on the Cardiovascular Risk

Contrary to theoretical assumptions and evidence from the general population, in which the role of DM as a cardiovascular risk factor is incontrovertible, studies that have evaluated the association between DM and cardiovascular disease in HeFH have offered contradictory results. Over the past two decades, a considerable number of studies have assessed the role of classical cardiovascular risk factors in patients with HeFH. A multi-centre retrospective cohort study performed in the Netherlands on 2400 patients (112,943 person-years) [153] found that, along with male gender, smoking, hypertension, low HDL-c and Lp(a), DM was independently associated with the presence of at least one cardiovascular event (RR 2.19; 95% CI: 1.36–3.54). Very recently, another methodologically similar study, which evaluated 1050 Japanese patients with HeFH over 19 years, also demonstrated that DM was an independent risk factor for a composite of major adverse cardiovascular events (HR 1.81; 95% CI: 1.12–2.25) [154]. However, the results of cross-sectional studies were mixed (see Table 3), and in many of them, DM was no longer significantly associated with the presence of cardiovascular disease after adjustment for other covariates. In many of the studies that found no association, either the population size was small or the prevalence of DM was very low, possibly limiting the statistical power to detect the association between DM and cardiovascular disease. In fact, a meta-analysis of 27 studies, published in 2018, aimed at assessing the association between cardiovascular disease and several classical risk factors, adding up to 41,831 subjects and 6629 cardiovascular events, found that DM was indeed an independent risk factor in HeFH (OR 1.95; 95% CI: 1.33–2.57), along with age, male sex, hypertension, body mass index, smoking, increased Lp(a), low HDL-c and a family history of cardiovascular disease [14].

In recent years, mainly due to the wide variation in established cardiovascular disease rates, even among individuals who share the same mutation and belong to the same family, there has been a growing interest in finding tools for cardiovascular risk stratification in subjects with HeFH. To this end, predictive models specifically designed for HeFH have been developed, and, strikingly, DM was not a factor to be taken into account in any of them. The first one, the Montreal-FH-SCORE, was calculated on the basis of retrospective data from a sample of 670 patients carrying a known FH-causing mutation in the *LDLR* gene, and it combines five predictor variables (age, gender, smoking, hypertension, and untreated HDL-c levels) [155]. In light of these findings, the authors conducted a specific study to investigate the impact of DM on cardiovascular disease in FH, using data from 1412 patients (73 with DM) from the FH Canada Registry. Although patients with DM had a higher prevalence of established cardiovascular disease, their results confirmed that including DM did not improve risk prediction with respect to the Montreal-FH-SCORE [146]. Subsequently, two mathematical models for cardiovascular risk prediction have been developed, but, unlike the Montreal-FH-SCORE, which had the limitation of being based on retrospective data, these were generated using prospective data from registries that collected incident cardiovascular events. The SAFEHEART Risk Equation was estimated using data from 2404 Spanish patients (104 with DM) with HeFH. Age, male sex, history of previous atherosclerotic cardiovascular disease, high blood pressure, increased body mass index, active smoking, and LDL-c and Lp(a) concentrations, but not DM, were independent predictors of incident cardiovascular events [156]. The FH-Risk SCORE was developed from a multinational prospective cohort of 3881 adults (152 with DM) with HeFH and no prior history of atherosclerotic cardiovascular disease. DM was not among the selected variables for the FH-Risk SCORE equation either, which incorporates sex, age, HDL-c, LDL-c, hypertension, smoking, and Lp(a) concentration as independent risk factors for 10-year atherosclerotic cardiovascular disease [157]. It should be noted that, until the publication of these two large studies, only a few long-term prospective studies had been carried out to assess the occurrence of new cardiovascular events in subjects with FH and, again, DM was not a significant risk factor in any of them [36,158,159].

Overall, the information available to date suggests that the role of DM as a cardiovascular risk factor in the FH population is smaller than in the general population. However, as their authors themselves acknowledge, due to the low prevalence among the FH population, even the highest quality prospective studies included small numbers of patients with DM and may not have had sufficient statistical power to determine the true effect of the disease [156,157]. Therefore, as has already been cautioned before [160], it is probably premature to underestimate the role of DM, and clinical judgement should be applied to establish the individual risk of a person with both FH and DM, considering other specific variables related to the disease, such as type of DM, time since diagnosis, or target organ damage, as recommended in clinical practice guidelines [161].

**Table 3 nutrients-14-01503-t003:** Cross-sectional studies that have assessed the association between diabetes and cardiovascular disease in subjects with heterozygous familial hypercholesterolemia.

Author, Year	Study Type *	Country	FH Diagnostic Criteria **	*N*	Diabetes (%)	Univariate Association OR (95% CI)	Multivariate Association OR (95% CI)	Adjusting Covariates
Hopkins, 2001 [162]	RR	USA	MEDPED criteria	262	3.0	NS	NS	Age, sex, BMI, smoking, waist to hip ratio, hypertension, HDL-c, triglycerides, small LDL, Lp(a), homocysteine, insulin, white cell count, C-reactive protein, xanthomas, intima-medial thickness, angiotensin-converting enzyme I/D polymorphism
De Sauvage, 2003 [163]	MC	Netherlands	Genetic test or definite DLCN criteria	526	2.1	17.61 (2.25–137.8)	NS	Age, sex, BMI, smoking, total-c, LDL-c, HDL-c, triglycerides, Lp(a), apo A1, apo B, homocysteine
Allard, 2014 [164]	SC	Canada	Definite DLCN criteria	409	6.4	3.2 (1.9–5.6)	3.6 (2.0–6.5)	Sex, BMI, smoking, family history of premature CVD, hypertension, LDL-c, HDL-c, triglycerides, Lp(a)
Alonso, 2014 [165]	MC	Spain	Genetic test	1960	3.9	Non reported	NS	Sex, BMI, smoking, hypertension, HDL-c, triglycerides, Lp(a), type of mutation, xanthomas
Besseling, 2014 [62]	NR	Netherlands	Genetic test	14,283	2.8	6.40 (5.21–7.86)	1.37 (1.03–1.82)	Age, sex, BMI, smoking, hypertension, lipid profile
Pereira, 2014 [166]	SC	Brazil	Definite or probable DLCN criteria	202	17.3	2.23 (1.05–4.75)	NS	Age, sex, BMI, smoking, hypertension, sedentary lifestyle, LDL-c, HDL-c, triglycerides, glucose, creatinine, xanthomas, corneal arcus, ankle-brachial index, claudication
Chan, 2015 [167]	SC	Australia	Genetic test	390	1.3	2.74 (1.06–7.08)	NS	Obesity, smoking, hypertension, CKD, LDL-c, HDL-c, triglycerides, Lp(a)
De Goma, 2016 [168]	NR	USA	Genetic test or any set of clinical criteria	1295	13	3.08 (2.04–4.64)	1.74 (1.08–2.82)	Age, smoking, hypertension, total-c, low HDL-c
Paquette, 2016 [155]	SC	Canada	Genetic test	670	3.3	3.5 (1.45–8.47)	NS	Age, sex, BMI, smoking, hypertension, prior statin use, total-c, LDL-c, HDL-c, triglycerides, VLDL-c, non-HDL-c, Lp(a), apoB
Paquette, 2017 [169]	MC	Canada	Genetic test	1388	4.5	3.28 (1.92–5.619	NS	Age, sex, BMI, smoking, hypertension, prior statin use, total-c, LDL-c, HDL-c, triglycerides, VLDL-c, non-HDL-c, Lp(a), apo B
Galema Boers, 2017 [170]	SC	Netherlands	Genetic test or definite or probable DLCN criteria	821	4	4.39 (2.15–8.97)	NS	Age, sex, BMI, smoking, hypertension, family history of CVD, previous cardiovascular disease, triglycerides, high LDL-c, low HDL-c.
Paquette, 2019 [146]	MC	Canada	Definite, probable or possible DLCN criteria	1412	5.2	2.9 (1.8–4.7)	NS	Montreal-FH-SCORE
Pérez-Calahorra, 2019 [171]	NR	Spain	Genetic test or definite or probable DLCN criteria	1958	6.5	4.99 (3.43–7.26)	NS	
Michikura, 2022 [172]	SC	Japan	Genetic test	176	12	Non reported	NS	Age, sex, BMI, smoking, hypertension, LDL-c, HDL-c, triglycerides, Achilles tendon elasticity index

* Type of study. SC: single-centre; MC: multicentre; RR: regional registry; NR: national registry. ** Diagnostic criteria. MEDPED: Make Early Diagnosis to Prevent Early Deaths System; DLCN: Dutch Lipid Clinic Network; NS: Not significant; BMI: body mass index; CVD: cardiovascular disease; c: cholesterol.

## 5. Knowledge Gaps and Further Research

The previous sections have highlighted the interplay between lipid and glucose metabolism, but also the controversy in this area. The inverse correlation between LDL-c concentrations and the risk of DM is supported by the low risk of DM in most populations with HF, by mendelian randomization studies, and by the increased risk of DM associated with some cholesterol-lowering agents, especially statins. However, results are inconsistent, and robust mechanistic studies are sparse. Furthermore, healthy behavior in people with FH could be associated with lower body mass index and a lower risk of T2DM.

There are several approaches that could fill in some of the existing knowledge gaps.
In FH populations where DM is more frequent than in the general population, family co-segregation studies could be performed, comparing the prevalence of DM and pre-DM in FH-causing mutation carriers and non-carriers in the same families;Studies focused on glucose tolerance, insulin secretion, and insulin resistance in whole-body and β-cell specific *LDLR* (or other FH-related genes) knock-out animal models, as performed already for *PCSK9* [129,130];FH-causing-mutation-specific studies in β-cells and islets, assessing their viability and function;Larger and longer prospective studies assessing the incidence of DM in FH and non-FH populations, as well as the cardiovascular risk of the combination of FH and DM.

## 6. Conclusions

Both DM and FH are associated with an increased risk of cardiovascular disease. Many studies suggest that FH is protective against the development of DM and that cholesterol-lowering treatments, especially statins, increase the risk of DM. Indeed, the LDLR is hypothesized to play a role in the toxicity of (or protection from) cholesterol on the β-cells. Their reduced amount or function in HF would protect the cells against LDL particle entry, whereas their increase would promote it and, thus, damage the β-cells. Nevertheless, this hypothesis is still to be proven. Indeed, a healthy lifestyle associated with a relatively low body mass index in people with FH could also account for some of the protection against DM. On the other hand, there are also studies showing an increased prevalence of DM in people with FH, and not all cholesterol-lowering drugs are associated with an increased risk of DM. The combination of FH and DM would be expected to be associated with an especially high risk of cardiovascular disease. However, existing evidence suggests that other classical cardiovascular risk factors modulate cardiovascular risk in FH, but DM does not play a highly relevant role. Short follow-up and small numbers of people with DM advise that this conclusion should be drawn with caution. Much research is still needed to fully understand the interplay between glucose and lipid metabolism in FH and DM.

## Figures and Tables

**Figure 1 nutrients-14-01503-f001:**
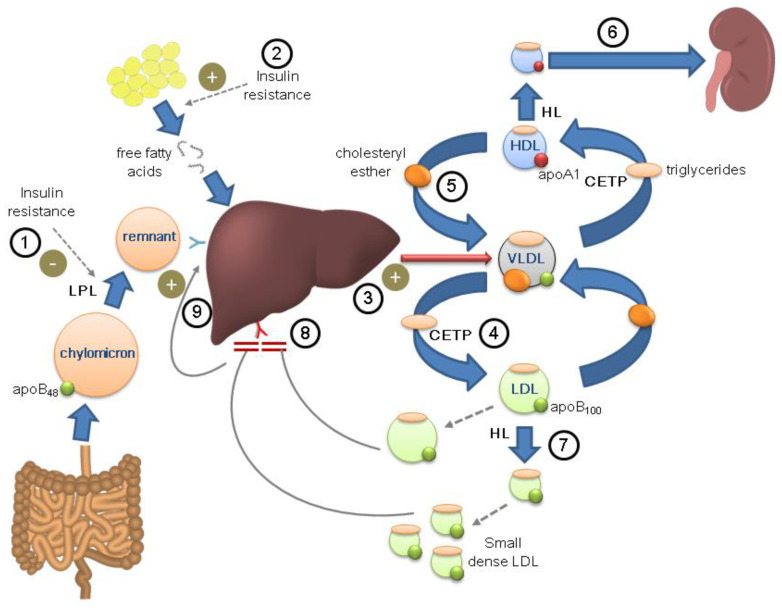
Potential combination of the physiopathological mechanisms of diabetes and familial hypercholesterolemia in the same individual. Diabetic dyslipidemia. Insulin resistance reduces lipoprotein lipase activity (LPL) ①, decreasing plasma triglyceride clearance, and promotes the release of free fatty acids ②, which are taken up by the liver and used for the synthesis and release of VLDL ③. VLDL exchange triglycerides and cholesterol esters with LDL ④ and HDL ⑤ through the action of cholesteryl ester transfer protein (CETP). Triglyceride-rich HDL particles, through the action of hepatic lipase (HL), are converted into smaller particles, with less anti-atherogenic properties, which are cleared more rapidly in the kidney ⑥. LDL particles also become smaller and denser (LDL phenotype B), more pro-atherogenic ⑦. Familial hypercholesterolemia. The genetic defect in LDL receptor prevents its uptake and metabolism in the liver, favoring the accumulation of LDL particles ⑧. This generates an increase in the uptake of chylomicrons and remnants in the liver ⑨, in turn boosting the synthesis of VLDL.

**Table 1 nutrients-14-01503-t001:** Prevalence of diabetes in representative populations with FH.

Author, Year	Country	*N*	Sample Characteristics	Diagnostic Criteria of FH	Diabetes (%)
Ferrières, 1995 [64]	Canada	263	French Canadian HeFH patients	Genetic test (*LDLR* mutation)	Men with CHD 1.9% Women and men without CHD 0%
Vuorio, 1997 [69]	Finland	179	55 HeFH with CHD and 124 HeFH without CHD	Genetic test (*LDLR* mutation)	9 and 0%, respectively
Neil, 1998 [63]	UK	1185	HeFH	Simon Broome Criteria	1.2% men 0.5% women
Fuentes, 2015 [70]	Spain	3823	2558 HeFH vs. 1265 unaffected relatives	Genetic test (*LDLR* mutation)	2.3%
Saavedra, 2015 [71]	Canada	188	HeFH	Genetic test (PCSK9-InsLEU or LDLR mutations)	4 and 2%, respectively
Besseling, 2015 [26]	Netherlands	63,320	25,137 HeFH vs. 38,183 unaffected relatives	Genetic test (*APOB, PCSK9* or *LDLR* mutations)	1.75%
Skoumas, 2017 [72]	Greece	280	90 HeFH vs. 112 familial combined hyperlipidemia vs. 78 controls	Clinical criteria or genetic test	2%
Climent, 2017 [65]	Spain	1732	HeFH	Definite or probable DLCN criteria	5.9%
Sun, 2018 [68]	China	289	HeFH	Definite or probable DLCN criteria	20.1%
Sánchez-Hernández, 2021 [66]	Spain	68	p.[Tyr400 Phe402del] *LDLR* carriers	Genetic test (*LDLR* mutation)	25%
Mehta, 2021 [73]	Mexico	336	332 HeFH and 4HoFH	Definite, probable, or possible DLCN criteria	11.3%

DM: diabetes mellitus, BMI: body max index, CHD: coronary heart disease, HeFH: Heterozygous familial hypercholesterolemia, HoFH: Homozygous familial hypercholesterolemia, DLCN: Dutch Lipid Clinical Network.

**Table 2 nutrients-14-01503-t002:** Studies assessing the association between lipid-lowering drugs and disorders of glucose metabolism.

Author, Year	*N*	Characteristics/Therapy	Mean Follow-Up	Mean Results	Statistical Measures (OR, HR or RR) (95% CI)
Sattar, 2010 [76]	91,140	Meta-analysis. All statins	4 years	NODM 9%	OR 1.09 (1.02–1.17)
Waters, 2013 [77]	15,056	Atorvastatin 80 mg vs. atorvastatin 10 mg or simvastatin 20–40 mg	4.9 years	0–1 NODM risk factors: NODM 3.22% vs. 3.35% 2–4 NODM risk factors: NODM 14.3% vs. 11.9%	HR 0.97 (0.77–1.22) HR 1.24 (1.08–1.42)
Cederberg, 2014 [78]	8749	Non-diabetic patients. All statins vs. control	5.9 years	NODM 11.2% vs. 5.8% High and low dose simvastatin High dose atorvastatin	HR 1.46 (1.22–1.74) HR 1.44 (1.23–1.68) and 1.28 (1.01–1.62) HR 1.37 (1.14–1.65)
Khan, 2019 [79]	163,688	Non-diabetic patients. Intensive therapy (PCSK9i or statins) vs. less intensive therapy (placebo/usual care)	4.2 years	NODM 6.1% vs. 5.8%	RR 1.07 (1.03–1.11)
Ko, 2019 [80]	2,162,119	Duration of statin use (<1 year vs. 1–2 years vs. >2 years) Cumulative dosing of statin (low-tertile vs. middle-tertile vs. high-tertile)	3.9 years	NODM 8.2% vs. 14.6% vs. 19.8% NODM 6.7% vs. 11.5% vs. 18.6%	HR 1.25 (1.21–1.28) vs. 2.22 (2.16–2.29) vs. 2.62 (2.56–2.67) HR 1.06 (1.02–1.10) vs. 1.74 (1.70–1.79) vs. 2.52 (2.47–2.57)
Choi, 2018 [81]	2483	5–10 mg rosuvastatin vs. 10–20 mg and atorvastatin vs. 2–4 mg pitavastatin	3 years	NODM 10.4% vs. 8.4% vs. 3%	HR Rosuvastatin vs. Pitavastatin: 3.9 (1.8–8.7) HR Atorvastatin vs. Pitavastatin: 2.6 (1.2–5.9)
Freeman, 2001 [82]	5974	All statins	3.5–6.1 years	NODM 2.3%	Pravastatin therapy HR 0.70 (0.50–0.99)
Hiramitsu, 2010 [83]	120	Ezetimibe	12 weeks	HbA1c: −3.4%; *p* = 0.05	
Dagli, 2007 [84]	100	High-dose pravastatin (40 mg) vs. combination low-dose pravastatin (10 mg) plus ezetimibe (10 mg)	6 months	HOMA IR: 3.16 vs. 2.05; *p* = 0.01	
Her, 2010 [85]	76	Atorvastatin 20 mg vs. rosuvastatin 10 mg vs. atorvastatin 5 mg plus ezetimibe 5 mg	8 weeks	HbA1c: +3% vs. +1.2% vs. −0.4%; *p* = 0.03	
Takeshita, 2013 [86]	32	Ezetimibe vs. placebo in NAFLD patients	6 months	HbA1c: 6.5% vs. 6%; *p* = 0.041	
Sabatine, 2017 [87]	27,564	EVOLOCUMAB vs. placebo	2.2 years	NODM 8% vs. 7.6%	HR 1.05 (0.94–1.17)
de Carvalho, 2017 [88]	68,123	Meta-analysis: PCSK9i vs. placebo	78 weeks	Mean difference in FBG 1.88 (0.91–2.68) mg/dL; *p* < 0.001 HbA1c 0.032% (0.011–0.050); *p* <0.001 NODM	RR 1.04 (0.96–1.13); *p* = 0.427
Chen, 2019 [89]	65,957	Meta-analysis: PCSK9i vs. placebo		Global NODM ALIROCUMAB Homogeneous statin use ALIROCUMAB and EVOLOCUMAB vs. ezetimibe	RR 0.97 (0.91–1.02) RR 0.91 (0.85–0.98) RR 2.14 (1.12–4.07) RR 0.60 (0.37–0.99)
Leiter, 2022 [90]	3621	Bempedoic acid vs. placebo	1 year	NODM 0.3% vs. 0.8%; *p* > 0.05 T2DM: HbA1c −0.12% vs. 0.07%; *p* < 0.0001 pre-T2DM: HbA1c −0.06% vs. −0.02; *p* < 0.0004	
Masson, 2020 [91]	3629	Meta-analysis: bempedoic acid vs. placebo	4–52 weeks	NODM	OR 0.66 (0.48–0.90)
Handelsma, 2010 [92]	216	Colesevelam vs. placebo in pre-T2DM patients	16 weeks	FBG: −4.0 mg/dL vs. −2.0 mg/dL; *p* = 0.02 HbA1c: −0.12% vs. −0.03%; *p* = 0.02	

OR: odd ratio; HR: hazard ratio; RR: risk ratio; CI: confidence interval; NODM: new-onset diabetes mellitus; HbA1c: glycosylated hemoglobin; HOMA-IR: insulin-resistance index; NAFLD: non-alcoholic fatty liver disease; PCSK9i: PCSK9 inhibitors; FBG: fasting blood glucose; T2DM: type 2 diabetes.

## Data Availability

Not applicable.

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
