# Peer review of "Diabetes and Familial Hypercholesterolemia: Interplay between Lipid and Glucose Metabolism"

_nutrients, 2022, doi:10.3390/nu14071503_

Round 1

Reviewer 1 Report

Diabetes and familial hypercholesterolemia: interplay between lipid and glucose metabolism is a well written review about the association between DM and familial hypercholesterolemia. My only suggestion is elucidating association between inflammation and those 2 conditions since both type 2 DM and dyslipidemia have close interaction with chronic, low grade inflammation. Neglecting serum uric in such a review would not be fair. Increased levels of serum uric acid have been reported both in type 2 diabetes mellitus and dyslipidemia. 

In summary, the review was very well written and just require above suggested revisions before reconsideration for publication. 

Author Response

Thank you for your comments.

A new section 4.3 been added describing inflammation and mentioning uric acid:

"Effects on chronic arterial wall inflammation and endothelial dysfunction"

Reviewer 2 Report

This manuscript summaries a large body of work on the molecular basis of the disease familial hypercholesterolemia (FH) and the potential interactions between FH and another metabolic disease, diabetes mellitus (DM).  The authors began by giving background on both FH and DM and then described the genes involved in the pathogenesis of these diseases.  The authors then went on to describe the results of specific studies showing that the risk of developing DM is altered in people with FH and those being treated with various lipid-lowering drugs, and the clinical consequences of the coexistence of FH and DM.  The manuscript flows well and is very organized; I think the way that the authors summarized the many different studies describing how different genetic causes of FH and various pharmacological treatments of FH are linked to the prevalence of DM and how that affects dyslipidemia connects many different studies together in a very way that makes sense to the reader and provides context.  The authors then end the manuscript well by providing some of the current gaps in knowledge and strategies to approach addressing these unknowns. 

I only have a few minor comments that I think would strengthen this manuscript even more:

1)The title for section 2 (Familial Hypercholesterolemia: molecular causes) could be broadened to include Diabetes Mellitus since this section discusses the genetic causes of both diseases.

2)While the authors did a nice job describing many studies where FH patients were treated with lipid-lowering drugs (section 3.2), I think it might be worth adding a table summarizing all these studies, like the summaries included in Tables 1 and 2.

3)The text on page 9 describing Figure 1 should be included in the Figure legend and not as text in the body of the manuscript.

4)In section 5 of the manuscript, the several approaches to address the gaps in knowledge that the authors include might make more sense as a number list instead of sentences in paragraph form.

Author Response

Thank you for your positive comments and for your pertinent suggestions

1) We have changed the title of section 2 to: 

"Familial Hypercholesterolemia and Diabetes: molecular causes"   2) A new table (now table 2) has been added, summarising the studies assessing the association between lipid-lowering treatments and glucose metabolism disorders   3)Thank you for spotting this. The text is now part of the figure legend   4)The approaches to fill in the knowledge gaps have now been numbered